# A Novel Tactile Sensing System Utilizing Magnetorheological Structures for Dynamic Contraction and Relaxation Motions

**DOI:** 10.3390/s23229035

**Published:** 2023-11-08

**Authors:** Yu-Jin Park, Bo-Gyu Kim, Eun-Sang Lee, Seung-Bok Choi

**Affiliations:** 1Korea Initiative for Fostering University of Research & Innovation, Inha University, Incheon 21999, Republic of Korea; eugene5059@inha.ac.kr (Y.-J.P.); leees@inha.ac.kr (E.-S.L.); 2Department of Mechanical Engineering, The State University of New York, Korea (SUNY Korea), Incheon 21985, Republic of Korea; 22192015@inha.edu; 3Department of Mechanical Engineering, Industrial University of Ho Chi Minh City (IUH), Ho Chi Minh City 70000, Vietnam

**Keywords:** magnetorheological fluid, magnetorheological elastomer, tactile device, dynamic motion, magnetic field, exciting frequency, viscoelastic, human tissue

## Abstract

It is well known that the rheological properties of magnetorheological (MR) material change under a magnetic field. So far, most works on MR materials have been oriented toward actuating characteristics instead of sensing functions. In this work, to realize dynamic tactile motion, a spherical MR structure was designed as a sensor, incorporating a magnetic circuit core to provide maximum dynamic motion. After manufacturing a prototype (sample), a sinusoidal magnetic field of varying exciting frequency and magnitude was applied to the sample, and the dynamic contraction and relaxation motion depending on the exciting magnetic field was observed. Among the test results, when 10% deformation occurred, the instantaneous force generated was from 2.8 N to 8.8 N, and the force when relaxed was from 1.2 N to 3.5 N. It is also shown that the repulsive force within this range can be implemented using an acceptable input current. The special tactile sensing structure proposed in this work can be used as a sensor to measure the field-dependent viscoelastic properties of human tissues such as stomach, liver, and overall body. In addition, it could be usefully applied to robot surgery, because it can mimic the dynamic motions of various human organs under various surgical conditions.

## 1. Introduction

Sensors have become indispensable entities in the era we live in, operating in diverse environmental situations. Most of the devices and machines we use incorporate various sensors to transmit different types of information. For those sensors used to measure human organs, visual and tactile information is the most conveyed. The transmission of tactile information occurs through various methods that consider how sensing is performed and how it is implemented. These transmissions respond to the characteristics of materials and surfaces, as well as the force of contact. Furthermore, there are various natural signals that can be collected depending on the external environment. These include mechanical, thermal, and chemical signals. In terms of tactile aspects, proprioception (force, motion) and haptic sensations (touch, contact) are the two most common and important types that can be collected [1]. Information related to tactile sensors can serve as an inspiration for creating virtual environments. In this context, various technologies have been integrated and presented. Many researchers have studied the ways to convey tactile information. The interaction of texture and haptic feedback has been explored in various interdisciplinary research fields. As research on tactile information transmission has progressed, it has found applications in many industries, including attempts in the medical field. One noteworthy aspect is the allocation of multiple functions within a single entity. As an example, Maingreaud et al. [2] created a small tactile map integrate in the hand to convey distance information to visually impaired individuals by integrating visual cues, sound, tactile sensations, and the surrounding environment. This small tactile device map was dynamically implemented using 64 micro coils. Paolo Motto et al. [3] conducted research on a dynamic tactile display based on piezoelectric technology—specifically, an 8 × 8 matrix of plastic pins to reconstruct braille, in order to aid visually impaired individuals. By identifying key parameters for tactile conversion in braille devices, a flexible device was designed to offer a high resolution and fast refresh rates, enabling braille rendering. M Pruzzini et al. [4] used electrorheological fluids to scan actual material samples, refining the data and generating electrical stimuli to reproduce roughness and texture sensations. They successfully replicated textural representations, including those of materials such as wood, paper, rubber, and fabrics. Kim et al. [5] conducted research on a pin-array-based solenoid actuator to provide detailed textures in a small-scale format. The proposed device demonstrated the capability to convey various tactile sensations with a broad frequency bandwidth. Furthermore, due to its extremely compact size, it shows the potential for implementation in various areas using multiple devices. Stanley [6] developed a variable-stiffness tactile display using pneumatic and particle jamming. This method involved creating chambers with a silicon layer and filling the interior with coffee grounds and empty spaces, controlling the stiffness by regulating the pneumatic pressure to manipulate tactile sensations. This offers the flexibility to generate various stiffness levels and surface shapes and to easily create arrays with multiple configurations, demonstrating potential as a dynamic model for providing tactile feedback. In addition, tactile devices have been researched in various forms, with applications spanning across diverse industries such as the medical field [7,8,9,10,11,12,13,14].

Recently, there have been efforts to integrate tactile sensors into surgical robots in the medical industry. Robots equipped with tactile sensing capabilities have been recognized as a relatively fundamental technology in the field of robotics and, hence, have also seen significant development outside of the medical domain. However, as of now, there is a lack of commercially available surgical robots with such functionalities. Tactile feedback systems are frequently criticized with regard to several aspects, such as the lack of leading and sufficient research efforts during the early stages of robotic surgery. Since all surgical robots operate based on visual surgical planes, the addition of tactile sensing capabilities to robotic surgery is anticipated to expand its applicability to a broader range of surgical procedures. Notably, there have been several cases where applications using magneto-rheological materials responsive to magnetic fields have been studied. For example, Kim et al. [8] conducted research proposing a tactile device that implements the proprioceptive sensation of actual organs using a single magnetorheological fluid-sponge cell. They aimed to apply the device in the medical field, and their study involved simulating robotic surgery to guide surgeons, fabricating the device, and adapting it to robots. To compare and validate the system, pig tissues were used, effectively demonstrating the ability to convey tactile information such as tissue resistance and repulsive force. Savioz et al. [15] designed and studied a small linear actuator using magnetorheological fluid with the goal of applying it to a tactile device and embedding it in a portable feedback glove. To ensure sufficient force in a compact device, a flow-mode piston type was adopted. It is noteworthy that this is a structure that can generate large forces using a small amount of magnetorheological fluid. Several works [16,17] analyzed the characteristics of an MR sponge and constructed a surgical robot system integrated with a 6-degree-of-freedom haptic master and slave to mimic several repulsive force characteristics of human-like organs. Then, a simple robotic incision surgery system was constructed by manufacturing a haptic master and slave robot for robot operation. The experimental results showed excellent control performance and high accuracy, with the tactile device allowing the operator to feel the rigidity of the surgical object.

As evident from the above literature survey, most of the systems for various tactile devices proposed so far have focused on texture and repulsive force. Studies on the shape changes and dynamic motion of tactile devices are considerably rarer, despite it being a very important issue in research fields such as robot surgery and artificial organs for people or animals. Human organs such as heart and stomach continuously move, and the movements change depending on the dynamic motions of the human being. In general, human movements are undertaken with contraction and relaxation in real time, like viscoelastic materials. In a special case, dynamic movements occur at the same time as changes in stiffness, which is related to the natural frequency of each human organ. Thus, an investigation of human organs in terms of dynamic motion is significant in order to avoid resonance and achieve high surgical accuracy. Therefore, in this study, a novel tactile device is developed, and its dynamic motions are investigated to identify its ability to mimic the dynamic motions of human organs such as the liver. After explaining the design and experimentation of devices capable of not only controlling the firmness of tactile sensations within manageable ranges but also inducing changes in shape, the dynamic motions are tested as functions of the exciting frequency and magnetic field intensity. It is clearly observed that the proposed tactile sensing structure demonstrates the possibility of changes in dynamic motions with differing amplitude. To the authors’ knowledge, there are no prior reports on tactile sensing devices utilizing MR structures that can generate shape changes and dynamic motions in real time. Consequently, this paper contributes to the literature by providing results of interest to many research fields, including sensors, robot surgery, and MR materials.

## 2. Characteristics of the Main Materials

In this study, the inherent characteristics of the main materials used to construct the structure for implementing motions of contraction and expansion, including magnetorheological fluid (MRF) and magnetorheological elastomer (MRE), are briefly described. These materials, which respond to magnetic fields, exhibit diverse properties and can be implemented according to the designer’s intentions, making them versatile for various fields of application.

### 2.1. Magnetorheological Fluid

MRF is a smart material that contains fine iron powder in an oil base such as silicone. When a higher yield stress is required, a higher viscosity and density of carbonyl iron powder (CIP) are used. There are various types of oil base and CIP. Commercial products distributed by Lord company in the United States are widely known. MRF is in a liquid form at room temperature (Figure 1a). As shown in Figure 1, it can be changed to a semi-solid state under the influence of a magnetic field (Figure 1b).

A semi-solidified fluid has a yield stress, and the value of this yield stress varies depending on the strength of the magnetic field. This principle is due to CIP particles in the fluid interacting and becoming magnetized. As the attractive force between CIP particles becomes stronger, the yield stress becomes stronger. As shown in Figure 2, the CIP in the MRF interacts along the magnetic field lines to form chains and become semi-solid. When these chains are released, the MRF becomes a fluid again and is restored to its original state.

This interaction becomes more robust as the magnetic field becomes stronger. The CIP forms tight chains and becomes thicker. Using these characteristics, the strength of the chains can be adjusted by adjusting the strength of the magnetic field. When placed under the influence of a magnetic field, the fluid becomes semi-solid and the yield stress changes. The yield stress becomes saturated when the chains are no longer strong. This allows us to determine the maximum control range. The MRF used in this study was Lord’s MRF-122EG (MRF-122EG, Parker Lord, Cary, NC, USA [18]). The main properties of the commercial products used are listed in Table 1.

### 2.2. Magnetorheological Elastomer

Magnetorheological elastomer (MRE) was used as the second material for designing the tactile structure proposed in this work. It was chosen as the material to implement a flexible and elastic form in the tactile device. MRE is also a kind of smart material created by mixing CIP particles into a base material like silicone rubber to generate interactions among the CIP particles under the influence of a magnetic field. MRE allows for stiffness adjustment under a magnetic field, and both the storage modulus and loss modulus are controlled by the magnetic field intensity.

Since there are no commercially available MRE materials, it must be manufactured directly. Silicone rubber (Smooth-on Company, Macungie, TX, USA, Ecoflex Silicone Rubber 0030: [19]) was selected as the base material for producing the magnetorheological elastomer. Considering the various hardness levels of silicone rubber, a flexible and stable material was required to mimic human skin. Therefore, silicone rubber with a low hardness level was chosen. This material is used to artificially replicate human skin in artificial organ models and special makeup applications. It was judged suitable for creating MRE when mixed with CIP due to its excellent elasticity and stability, allowing for recovery when subjected to external forces or removal. Additionally, the selected material, despite its low hardness, did not exhibit stickiness after hardening, making it easy to mix with CIP during production. Furthermore, it had a short curing time of 4 h at room temperature, which was advantageous for the production process.

The MRE was produced via the following steps:(1)Material Selection: In this research, the chosen base material, Ecoflex 0030 silicone rubber, was manufactured with a 1:1 ratio of base and curing agent.(2)Mixing: CIP from BASF in Germany was used. The model used was CIP CC (CIP CC, Basf, Manheim, Germany). The particle size of the CIP was 3.8–5.3 microns. Mixing should be completed within a short pot time, approximately 30 min, to prevent the CIP particles from clustering. Pre-measured CIP particles were added without clumping, using a skimmer to ensure an even blend. A simple vacuum experiment device was used to remove any bubbles generated during mixing. Using a simple pump, the mixed MRE was placed inside the experimental device within the pot time.(3)Molding: The mixture was poured into a prepared mold. Such molds are typically created using a 3D printer.(4)Curing: The mixture was allowed to cure at room temperature for approximately 4 h.

The method for producing MRE using the selected base material and CIP is illustrated in Figure 3. The use of 3D printed molds is common in the production of MRE.

An MRE was produced by mixing CIP at a 50 wt% ratio with base silicone rubber. Typically, a fraction of 70 wt% produces the best MR effect [20,21,22]. However, in this experiment, it was expected that there would be a risk of detachment from the base material if a large amount of CIP was used because the MRE was to be repeatedly deformed. After manufacture of the MRE, it was confirmed via rheology testing that its MR effect was continuously and sufficiently acceptable with 50 wt% CIP. An Anton Paar MCR-302 (Anton Paar Company, Graz, Austria MCR 320 model, Molecular Compact Rheometer [23]) was used to measure the properties of the produced MRE. The experiment was conducted using the Strain Amplitude Sweep Method. The experimental sample was 1 mm thick and 2 cm in diameter. The material was transformed via shear, and the resistance of the material generated as the strain increased to a certain rate was measured. The energy generated at this time was calculated, and the storage modulus and loss modulus were measured. Figure 4 shows an internal view of the measurement equipment used in the experiment and the device operation and configuration. The storage modulus and loss modulus were measured for each magnetic field strength.

The characteristic test results of the manufactured MRE are shown in Figure 5. The storage modulus is associated with the elastic component of the material (stored energy), while the loss modulus is associated with the viscous component (dissipated energy). The modulus value gradually increases for each magnetic field strength. As a reliable section, we refer to the modulus value near 0.1%, where it shows stability. As the magnetic field increases, the rate of increase in the storage and loss moduli increases. The data corresponding to the 0.1 T interval are shown in Table 2.

The modulus seemed to increase linearly, but the modulus increased significantly from the fourth section with a nonlinearity that was difficult to predict. It was confirmed that the modulus characteristics of the MRE changed depending on the magnetic field. The target of this study was to use MRF and MRE together. To determine what kind of effect it would have when MRE was used together with MRF, we magnetized them together and observed them using a scanning electron microscope. To prepare for this, an additional magnetization device was prepared. To capture the zone of the surface of the MRE and the shape of the CIP chains extending within the MRF base oil, a magnetization time of about 2 h was given before producing the SEM specimen, and post-treatment of the surface coating of the specimen was performed. Figure 6 shows that the CIP chains within the MRF extended from the surface zone of the MRE according to the direction of the magnetic field. The change in modulus under the influence of the magnetic field on the MRE was basically confirmed. In addition, a tactile structure was designed, manufactured, simulated, tested, and verified with the expectation that it could contribute to dynamic shape change under the influence of the magnetic field on the MRF.

## 3. Design Concept and Magnetic Analysis

### 3.1. Tactile Structure Concept

The proposed tactile structure changes its shape and state under the influence of a magnetic field as shown in Figure 7 below. There is a shape and hardness change area section up to a certain point, and there is a recovery section where everything returns to its original form after the magnetic field influence reaches its maximum point. To realize these points, the use of materials with elasticity and yield stress was introduced earlier. The outside is composed of MRE made of materials with good elasticity and recovery, and the inside is filled with MRF of low viscosity. The top part of the MRE is filled to excess so that it swells to a slightly convex shape. Manufacturing this structure is important, but the core part where the magnetic circuit is formed is also very important. Looking at Figure 7, the magnetic circuit is structured to rise from the bottom and fall outward on both sides. This is the concept of inducing a change in shape by pulling the initially convex upward shape downward by pulling strongly from below and then pulling it again to the side. To achieve this, an electromagnetic device that can pass a large magnetic field is needed to reliably pass through the tactile structure. For this purpose, we designed a new magnetic core unit and conducted magnetic field simulations and MATLAB simulations.

### 3.2. Magnetic Core Design

An electromagnetic structure must be formed to shape the magnetic field direction to maximize the contraction and relaxation of the manufactured tactile structure. Therefore, magnetic distribution analysis should be carried out as a first step. Figure 8 presents the proposed tactile structure associated with the magnetic circuit, which affects the deformation of the tactile structure by pulling and contracting. It is noted that a downward pulling force and a pulling force on both sides occur simultaneously in the tactile structure, allowing significant deformation to occur.

Magnetic field simulation was conducted using the same model of the core structure and the manufactured tactile structure. To obtain the theoretical value, the magnetic intensity applied to the tactile structure was calculated as follows.
(1)$$H=\frac{N_{c}\cdot i}{l_{TS}+\mu_{TS}\cdot A_{TS}\left(\frac{\frac{\mu_{2}\cdot A_{2}}{l_{2}}+\frac{\mu_{3}\cdot A_{3}}{l_{3}}\cdots \frac{\mu_{7}\cdot A_{7}}{l_{7}}+\frac{\mu_{8}\cdot A_{8}}{l_{8}}}{2}\right)}$$

In the above,H is the magnetic intensity$,\;\;N_{c}$ is the number of turns of the coil, and i  is the applied current. The permeability of MRE used a static value of 3. And, MRF used nonlinear B-H data. $\mu_{x}$ is the relative permeability of each part. The material of the core was 1008 steel, that of the bobbin was aluminum alloy, and that of the coil was copper. $A_{x}$ is the area through which the magnetic circuit passes, $l_{x}$ is the length of each part, and ‘ts’ is an abbreviation for tactile structure. When the simple calculation was performed using MATLAB, information on each core part was entered and a simulation was performed. When 1A was input, the magnetic field strength applied to the tactile structure was approximately 272.73 A/m. A magnetic analysis was performed to obtain visual confirmation and a transient simulation. The software used was Maxwell from Ansys Software (2019 version). The simulation results were post-processed and are shown in Figure 9 and Figure 10, displaying the magnetic field flow throughout the entire shape. The tactile structure was affected by the magnetic intensity between 0.2 and 0.8 kA/m. In general, the magnetic circuit was connected directly to the ferromagnetic material next to the tactile structure. The simulation was automeshed (automatically creates a mesh) based on a tetrahedral shape. The total number of nodes was 54,917, and because the shape was simple, the mesh was not set more densely. The entered number of turns of the coil was 3000, and for relative permeability, engineering data in Maxwell was used for all materials except MRE and MRF. The current was applied transiently up to 1A until it reached 10. The nonlinear residual value for convergence was entered as 10^−6^.

To induce the maximum deformation of the tactile structure, the magnetic field enters from below and pulls as much as possible from below; then, the magnetic field exits on both sides to induce deformation of the tactile structure. Through this, the structure is compressed downward. When the magnetic field is released, the original shape is restored, and the tactile structure that was pulled down and contracted is relaxed. In this study, the core design and the path of the magnetic circuit are important points. Figure 10 shows the strength of the magnetic field applied to the tactile structure through the contour. The pulling force from below and the shape distributed on both sides show that the design was carried out as intended.

## 4. Tactile Structure and Experimental Setup

The outer cover of the MRE structure, which was manufactured by blending silicone rubber and CIP and then curing it using a specially made 3D printed mold, is shown in Figure 11. The shape of the tactile structure is a rectangular parallelepiped measuring 2.5 × 2.5 × 1.5 cm^3^. The calculated internal volume is 9.375 mL. Figure 11a shows the external shape. In Figure 11b, you can see that the inside is empty. The thickness of this cover is 1 mm. Figure 11c shows a flat-type MRE used as a cover to fill and seal in the MRF. Like the molded cover, it is 1 mm in thickness. Figure 11d shows a silicone-only adhesive (Smooth-on Company, Macungie, TX, USA, sil-poxy [23])**.**

The inside of the manufactured MRE cover has a volume of 9.375 mL. To achieve pre-tension, 9.7 mL of MRF, which was an excess amount, was injected. First, before injection, the lower part was sealed using the prepared flat-type MRE, and the MRF was injected using a syringe in an inverted state. The injection area was even sealed with adhesive to prevent the MRF from leaking out. Figure 12 shows the proposed tactile structure in which MRF was injected into the MRE cover, which was then sealed and shaped. An amount exceeding the internal volume was injected, resulting in a convex shape in the upper and side parts.

The experimental setup for the repulsive force testing and dynamic shape change testing of the manufactured tactile structure is shown in Figure 13. Two experiments were conducted. First, a stress relaxation test was conducted to measure data on repulsive force. This is an experiment that measures the change in stress of the tactile structure under a certain strain depending on the magnetic field strength. A force gauge was used to measure the repulsive force, and the unit was Newtons. The compression ratio was 10%. The displacement was measured using a laser sensor. The motor drive stand was controlled using a computer to specify the degree of compression. Since the first test was an experiment in which the magnetic field strength was given statically, a total of five types of experiments were conducted within the current range from 0 to 1.23 A at intervals of 0.31A.

In the second experiment, the magnetic field was given as a frequency type and a test was conducted to repeatedly insert and remove the magnetic field applied to the tactile structure. Through this test, the change in shape of the tactile structure depending on the strength of the magnetic field was measured. In this experiment, a function generator to generate frequency was connected to a current amplifier, which was connected to an electromagnet with a core designed for the tactile structure. A current set at a frequency of 0.1 Hz, 0.5 Hz, or 1 Hz was input to this electromagnet. To record the phenomenon of repeated contraction and relaxation, a laser sensor was placed at the top of the tactile structure, and the height change was measured. All measured data were collected through a data collector and directly transmitted to a computer.

## 5. Results and Discussion

The stress relaxation test results are shown in Figure 14. The instantaneous force is shown to soar across all currents and then converge to a constant value. It can be seen that the field-dependent repulsive force measured for the proposed tactile structure exhibits stress relaxation behavior, which is a typical property of viscoelastic behavior. The most notable feature is that as long as the current is continuously supplied, it does not fall below a certain value. When compressed to about 10% of the maximum peak, this tactile structure can be viewed as having a control range of 2.8~8.8 N based on the peak value and 1.2~3.5 N based on the relaxation value under the control of the maximum current that can be applied.

The results of the second experiment are visually displayed in Figure 15. The photo frames show the tactile cycle changing under the influence of the magnetic field during a 1/2 frequency cycle. As the intensity of the magnetic field density rises to its peak, the downward and sideways pulling force becomes stronger, causing maximum shape change. In the section where the strength of the magnetic field weakens again, the original shape is restored as it enters the recovery zone. The shape of the graph above shows that stress builds over time because the tactile structure has viscoelastic properties. This can be expressed as the relaxation stress, *σ*(*t*). Because it is a time-dependent stress curve, the stress relaxation can be expressed by the Maxwell model as shown in Equation (2). It has a similar pattern to the creep phenomenon. $\sigma_{0}\;$refers to the peak value, which is a constant stress. θ represents the viscosity of the tactile structure, and *E* denotes the stiffness.
(2)$$\sigma \left(t\right)=\sigma_{0}\exp\left(-\frac{t}{\tau }\right),\;\tau =\frac{\eta }{E}=relaxation\;time$$

For the next test, the displacement was tested with input currents at 0.1 Hz, 0.5 Hz, and 1 Hz, and displacement data were measured at the top of the tactile structure using a laser sensor. All experiments were conducted in a total of 10 stages of testing from 0.12 A to 1.23 A at 0.13 A intervals. It was observed that as the current application increased, the displacement increased at the interval. The experimental results are shown in Figure 16. It can be seen from the graph that the amount of deformation varied depending on the applied frequency.

To collect data, we set the center line as shown in Figure 17. The collected data are shown in Table 3, where the unit is mm. The reason the measured data are negative is because the displacement that occurs as the tactile structure contracts downward was measured from an initial value of 0. In other words, it is a displacement that occurs downward from the center of the reference axis; hence, it is expressed as a negative value. Comparing the data in Table 3 confirms that the displacement amount is different for each frequency even though the current value is the same. This is because the tactile structure contracts and relaxes and cannot keep up with the speed at which its shape changes. At a relatively low-frequency input current, the shape has enough time to deform, so a large amount of deformation can occur. Conversely, at a high-frequency input current such as 1 Hz, there is not enough time for the shape to deform. In this phenomenon, the chains within the MRF are formed and the magnetic field is released again as they settle into the MRE cover, causing the contraction to continue for a certain period before returning to its original state. Thus, the response time of the MRF is very important to obtaining fast dynamic motion of the proposed tactile structure.

## 6. Conclusions

In this study, we fabricated a new type of tactile structure that can generate dynamic motion, exhibiting field-dependent repulsive force performance and shape changes across frequency bands. The proposed tactile structure consists of two different magnetically responsive materials. The MRE cover is made of a very flexible material, and overcharged MRF is sealed within to form a single cell; this structure was designed to be affected by magnetic fields while pre-tensioned. The magnetic core part was newly designed so that the tactile structure would be sufficiently affected by magnetic fields. An electromagnet device in a form that could accommodate the specially designed tactile structure was produced. The core was designed such that the magnetic field passes through the lower part of the tactile structure and exits to the side, creating a force that can pull the tactile structure laterally. While designing the core part, the magnetic intensity applied to the tactile structure was calculated and verified through magnetic simulation. Afterwards, the controllable range of the repulsive force was confirmed through a stress relaxation test under a magnetic field of uniform distribution. This test was conducted to check how well this structure performs in terms of repulsive force, which is a basic requirement for a tactile structure. When 10% deformation occurred, the instantaneous force generated was from 2.8 N to 8.8 N. The force when relaxed was measured to be from 1.2 N to 3.5 N. It was shown that repulsive force within this range can be implemented using the input current. In the second experiment, the frequency-type input current dynamic motion test was carried out. A magnetic field was continuously applied at a frequency type to induce shape changes in the tactile structure and measure displacement. Through this experiment, it was confirmed that the displacement was different at each frequency due to differences in the ability of the structure to contract and relax under the magnetic field depending on the applied speed. (It is noted here that a video showing the dynamic motion of the proposed tactile device has also been submitted as a Appendix A.)

It is finally remarked that the results obtained from this work are preliminary. The proposed results demonstrate the possibility that tactile structures can undergo dynamic motion simultaneously with changes in stiffness. Therefore, for successful dynamic object development, the following tasks must be completed in future: (1) Various shapes of the structure should be tried. (2) Optimization should be performed to reduce the required power. (3) Sealing issues to prevent the leakage of MRF should be resolved for guaranteed durability. In addition, it is noted that the final goal of this research is to control the shape and dynamic frequency of the tactile sensor for application to robotic surgery in which human organs exhibit different shapes and time-varying frequency. This can be accomplished using a proper feedback control system to adapt to unevenly structured organs and environmental-dependent frequency. Unevenly structured organs can be simulated by applying the control magnetic field to the magnetic cores of the proposed tactile sensor. This is practically realizable since field-dependent contraction and relaxation of the proposed tactile sensor can be generated by controlling the magnetic input field.

## Figures and Tables

**Figure 1 sensors-23-09035-f001:**
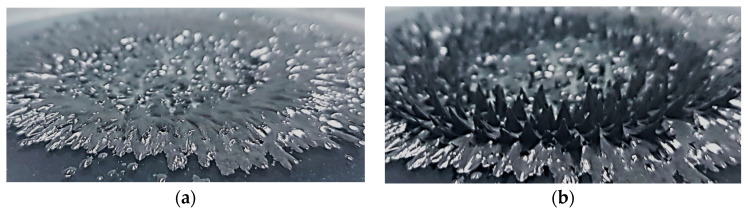
MRF state change with and without a magnetic field: (**a**) without a magnetic field; (**b**) with a magnetic field.

**Figure 2 sensors-23-09035-f002:**
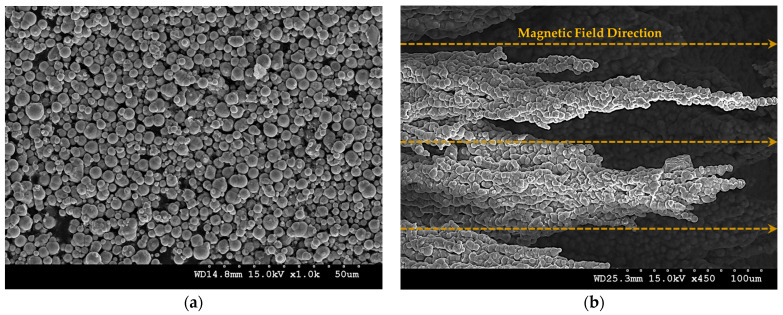
The micro-behavior of CIP in MRF: (**a**) without a magnetic field; (**b**) with a magnetic field.

**Figure 3 sensors-23-09035-f003:**
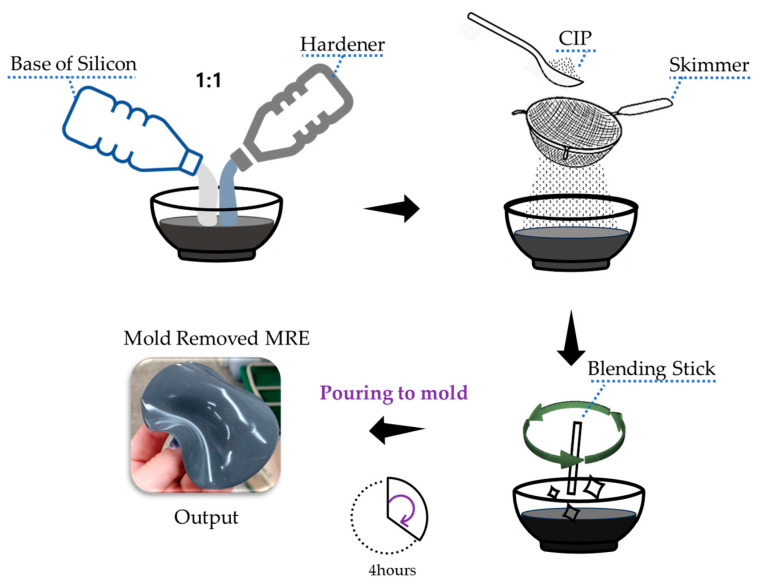
MRE manufacturing sequence.

**Figure 4 sensors-23-09035-f004:**
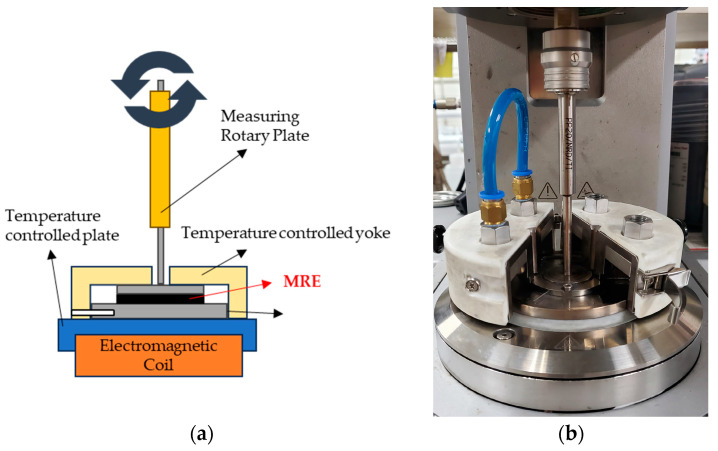
Rheology test equipment: (**a**) configuration; (**b**) photo showing an internal view.

**Figure 5 sensors-23-09035-f005:**
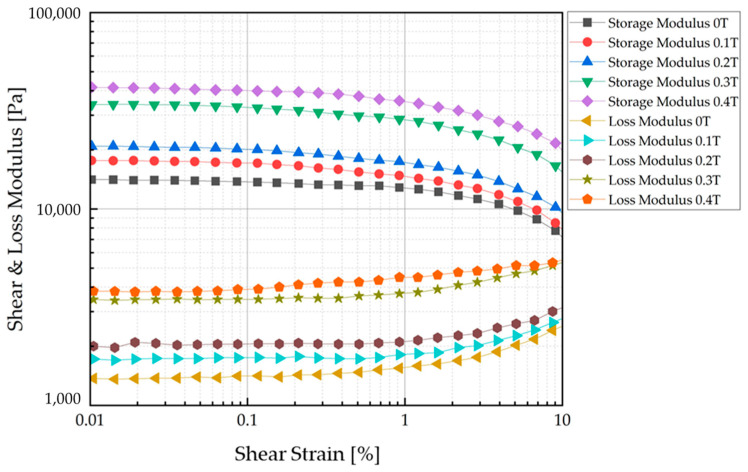
Strain amplitude experiment results: storage and loss moduli.

**Figure 6 sensors-23-09035-f006:**
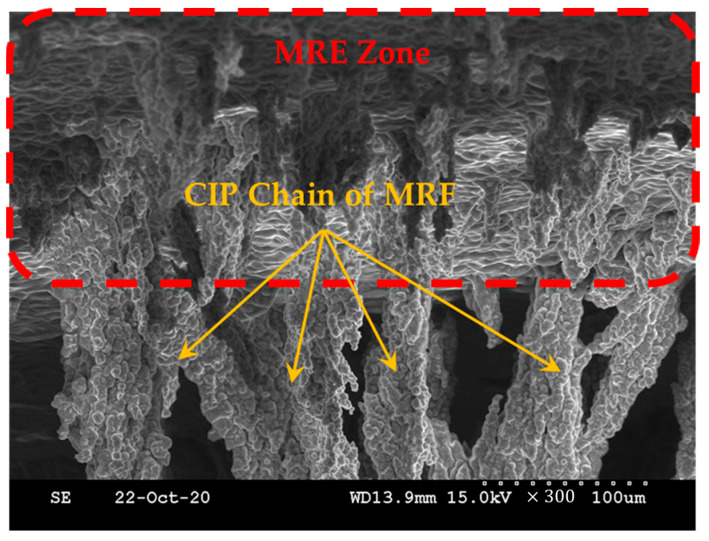
The shape of the MRF chains extending from the MRE zone.

**Figure 7 sensors-23-09035-f007:**
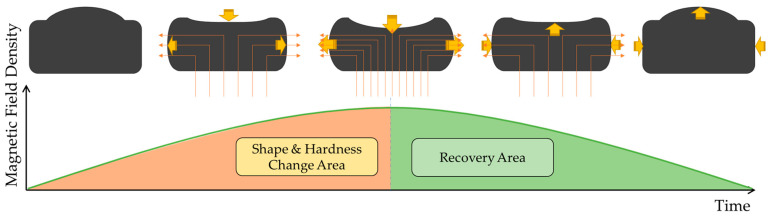
Concept diagram for the tactile structure’s shape change.

**Figure 8 sensors-23-09035-f008:**
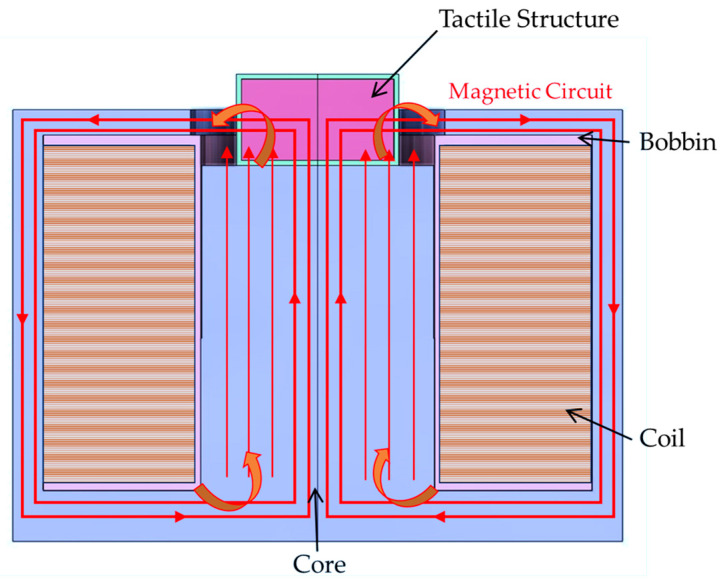
Tactile structure with an electromagnetic circuit.

**Figure 9 sensors-23-09035-f009:**
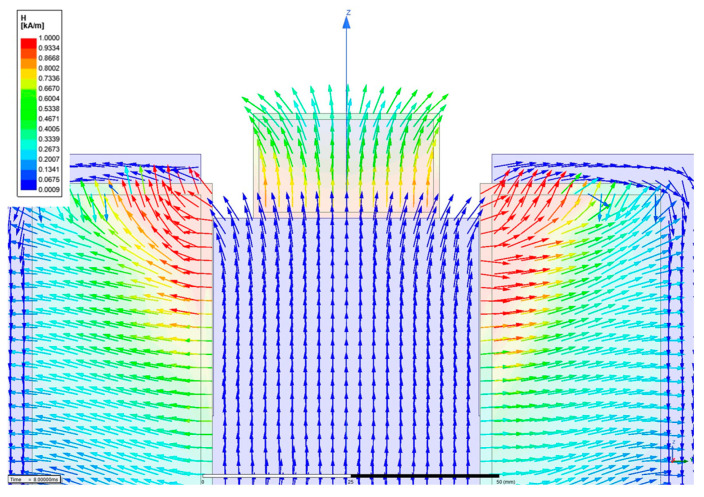
Magnetic simulation of the overall structure.

**Figure 10 sensors-23-09035-f010:**
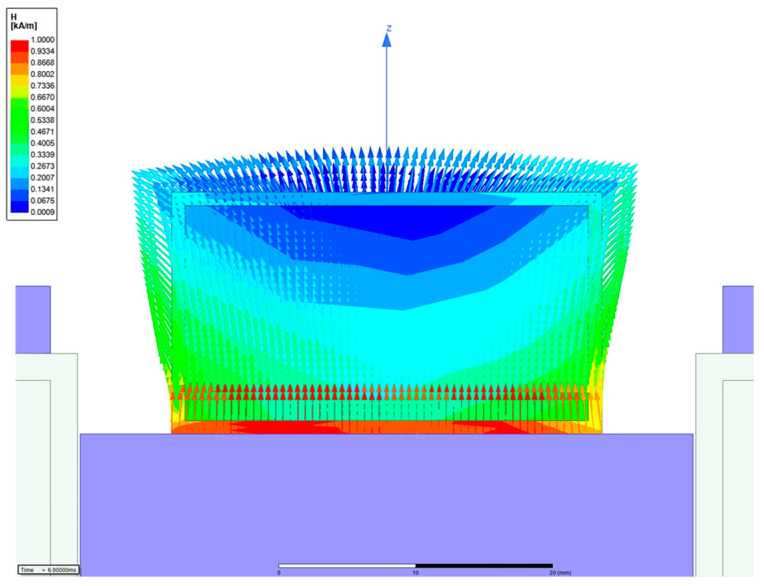
Magnetic simulation of the tactile structure (contour type).

**Figure 11 sensors-23-09035-f011:**
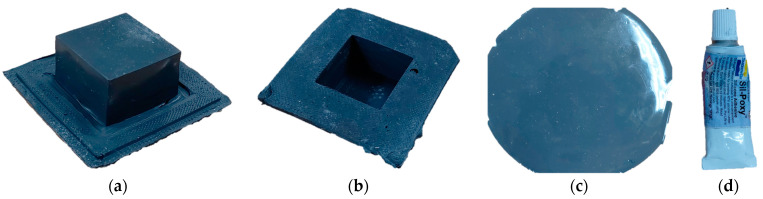
Materials for manufactured MRE covers and seals: (**a**) MRE cover (upper view); (**b**) MRE cover (lower view); (**c**) flat-type MRE; (**d**) adhesive for sealing.

**Figure 12 sensors-23-09035-f012:**
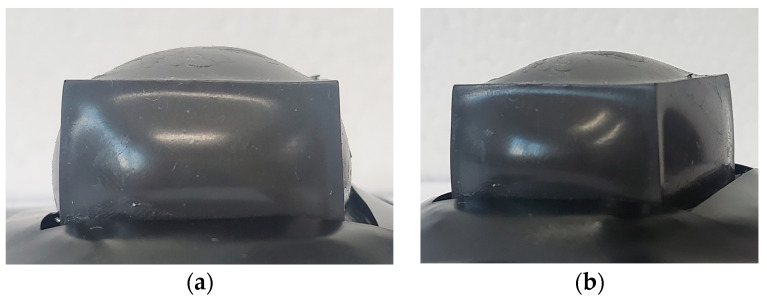
MRF filled and sealed inside the MRE cover: (**a**) front view; (**b**) overall shape.

**Figure 13 sensors-23-09035-f013:**
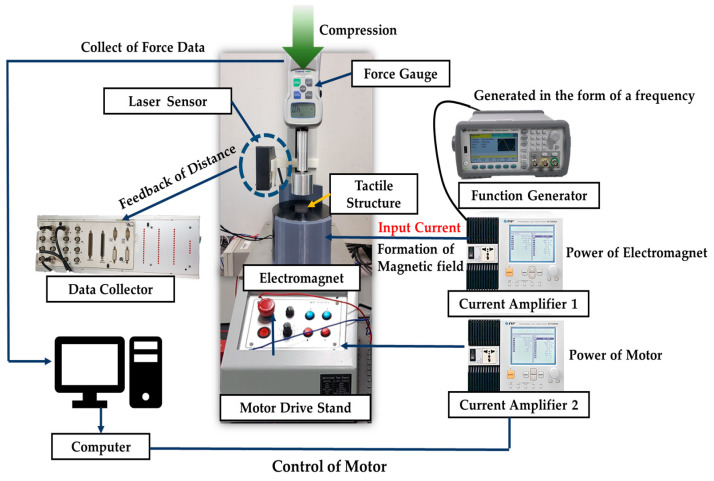
An experimental schematic for dynamic motion testing.

**Figure 14 sensors-23-09035-f014:**
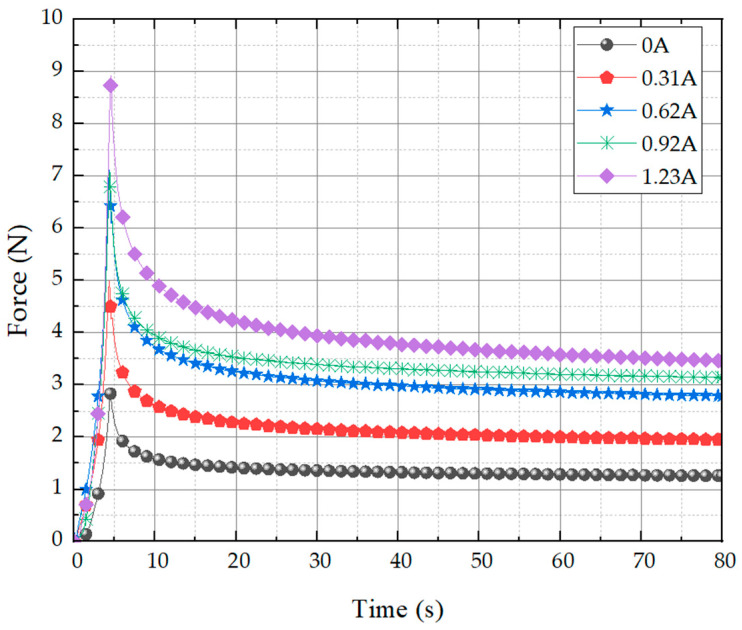
Stress relaxation of the proposed tactile structure.

**Figure 15 sensors-23-09035-f015:**
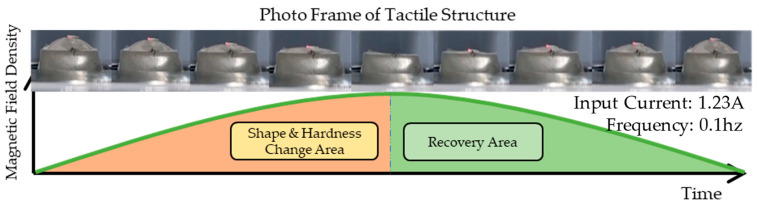
The tactile structure’s shape change under the influence of a magnetic field during a 1/2 frequency cycle (please refer to the video for this motion submitted as a Appendix A).

**Figure 16 sensors-23-09035-f016:**
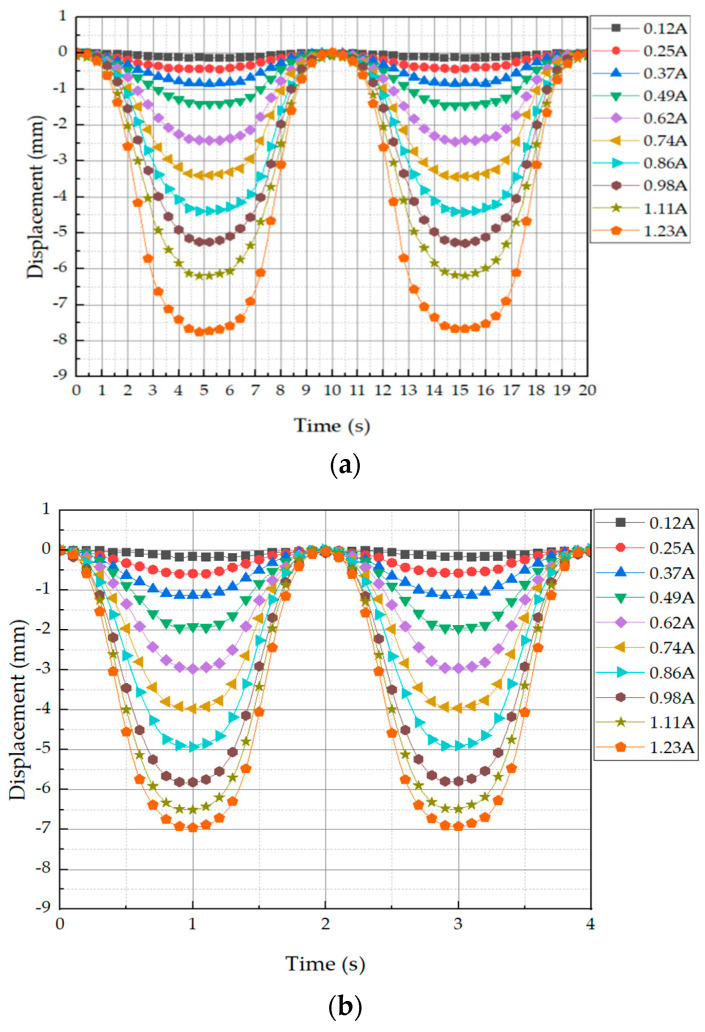
Height displacement measurement data of the tactile structure transformed by applying current in the form of frequency: (**a**) 0.1 Hz input current; (**b**) 0.5 Hz input current; (**c**) 1 Hz input current.

**Figure 17 sensors-23-09035-f017:**
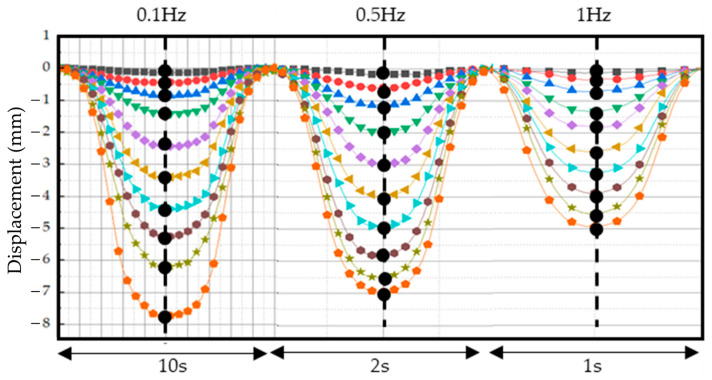
The location of the points by establishing a center line to collect data for each frequency.

**Table 1 sensors-23-09035-t001:** Properties of MRF-122EG, a Lord Corporation Product.

	(122EG)	(132DG)	(140CG)
Solid content by weight (%)	72	80.98	85.44
Density	2.28–2.48	2.95–3.15	3.54–3.74
Yield stress (200 kAmp/m)	32 kPa	44 kPa	60 kPa
Viscosity (Pa-s)	0.042 ± 0.020	0.112 ± 0.02	0.280 ± 0.070

**Table 2 sensors-23-09035-t002:** MRE 50 wt% rheology test results.

	0 T	0.1 T	0.2 T	0.3 T	0.4 T
Storage modulus	13,741 Pa	17,136.4 Pa	20,128.3 Pa	32,902.9 Pa	40,080.7 Pa
Loss modulus	1412.1 Pa	1752.1 Pa	1752.1 Pa	3467.9 Pa	3898.4 Pa

**Table 3 sensors-23-09035-t003:** Data values collected from the center line; unit: mm.

	0.1 Hz	0.5 Hz	1 Hz
0.12 A	−0.13	−0.15	−0.12
0.25 A	−0.42	−0.56	−0.34
0.37 A	−0.82	−1.11	−0.69
0.49 A	−1.49	−1.95	−1.32
0.62 A	−2.48	−2.96	−1.84
0.74 A	−3.47	−3.97	−2.61
0.86 A	−4.42	−4.89	−3.23
0.98 A	−5.33	−5.77	−3.87
1.11 A	−6.16	−6.44	−4.50
1.23 A	−7.70	−6.91	−4.92

## Data Availability

All data are available upon request, and a video showing the dynamic motion with shape change due to the exciting frequency has been submitted as a Appendix A.

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
