# Peer review of "A Novel Tactile Sensing System Utilizing Magnetorheological Structures for Dynamic Contraction and Relaxation Motions"

_sensors, 2023, doi:10.3390/s23229035_

Round 1
Reviewer 1 Report
Comments and Suggestions for Authors
The manuscript presents an interesting topic in the field of magnetorheology and its applications. I have some comments about the materials and methods used, and for the results:
(1) In section 2 of materials, are the photos in figures 1 and 2 your own or from literature?
(2) Could you include the viscosity value in table 1 for the MRFs?
(3) Provider of CIPs for the MRE? Particle size for CIPs?
(4) Line 234: how was the formation of bubbles or defects avoided?
(5) Why was 50%wt CIPs chosen for the MRE?
(6) From lines 252-258 text is repeated in the paragraph.
(7) Lines 262,263: Storage modulus is associated with the elastic component of the material (stored energy), while loss modulus with the viscous component (dissipated energy).
(8) The axes in figure 5 should be "Storage & Loss modulus" vs "Shear strain".
(9) I understand that strain sweep amplitude aims to determine the viscoelastic linear region..... why only that rheological measurement? At what temperature and frequency was the measurement carried out?
(10) I consider that a frequency sweep would also be important for this work (see results from figure 16).
(11) How did you measure the intensity of the magnetic field in the rheometer?
(12) Why was MRF magnetorheology not included?
(13) Check the text of lines 310-330, there are repeated phrases.
(14) Information is missing about the design and materials of the magnetic circuit.
(15) For equation 1, it is not explained in the text what LTS and uTS are.
(16) The simulation results, figures 9 and 10, are interesting, but more information is required in this regard: number of nodes, mesh convergence, calibration, etc.
(17) Line 450: displacement and deformation are not the same.
(18) From figure 14, it would be interesting to calculate the response time or relaxation time. A viscoelastic model can help with the control of actuation properties.
(19) The results shown (figures 15, 16 and 17) correspond to the behavior of an actuator. I do not understand or do not directly see the sensing function, as stipulated in the objective of the work.
(20) Some typos are found in the text. Please revise.
Comments on the Quality of English LanguageA grammar check is recommended.
Reviewer 2 Report
Comments and Suggestions for Authors
In this research, the author proposed a core-shell structure based on Magnetorheological Elastomer (MRE) as the shell and Magnetorheological Fluid (MRF) as the core. The study investigates the structural changes in the presence of an applied magnetic field. This structure may be the first of its kind proposed for use as a tactile sensor. The motivation of the paper is very interesting because this type of tactile sensor can be useful for high accuracy in the robotic surgery. The paper is well-written, and I can support its publication with some minor corrections:
1. The Introduction section is excessively lengthy and can be shortened.
2. Punctuation and other mistakes should be checked carefully.
3. Lines 310-330 should be deleted.
4. Scale bars for Figure 2 are necessary.
5. The authors have studied a specific-shaped tactile sensor. However, human organs have various shapes. Please introduce how this sensor can be beneficial for surgery on unevenly structured organs.
Round 2
Reviewer 1 Report
Comments and Suggestions for Authors
The comments have been taken into account, and the authors have made an effort to improve the quality of the manuscript.
Author Response
Authors’ Reply
We have reduced the self-citation as below.
1st Revision: No. of self-citation=7
2nd Revision: No of self-citation=2 [16] [21]
We have carefully checked all references, especially the changed references, are closely related to the main content of this work.